# Research on enterprise network public opinion guiding decision-making considering crisis differentiation

Jiakun Wang[1,2]*, Yawei Li[1], Xiaotong Guo[1], Huiying Chen[1]

1 College of Economics and Management, Shandong University of Science and Technology, 2 College of Computer Science and Engineering, Shandong University of Science and Technology, Qingdao China

* shandongwjk@163.com

## Abstract

### Background

The new media environment driven by digital intelligence technologies provides new opportunities for enterprise development, but also brings new challenges for enterprise crisis management and network public opinion information (referred as public opinion) guidance. Focusing on the contradictions between the complex diversity of crisis types and the limited governance resources, it is of great significance for enterprises to determine the best public opinion guidance strategy.

### Methods

Considering the dynamic complexity of public opinion derived from enterprise crisis, this paper innovatively proposed a decision-making model of enterprise public opinion under crisis differentiation based on differential game. Then, the balance strategy of each stakeholder and the guiding effect of public opinion were discussed under four decision scenarios. Finally, the optimal resources allocation ratio of different types of crises is determined under the constraints of governance resources, and the key parameters of public opinion guidance process are identified.

### Results

The results show that, under four decision scenarios, the dual-guidance strategy achieves the best public opinion guidance effect and Pareto-optimal outcome for the game system. The overall benefit is maximized when the optimal investment ratio for the four types of crises (Values-type, Product-type, Marketing-type, and Internal management-type) is 60:28:8.4:3.6. The subsidy coefficients of enterprise to netizens and media both significantly the guidance effect of public opinion, but compared with the former, it is more sensitive to the changes of the letter.

**Data availability statement:** All relevant data are within the manuscript and its Supporting Information files (named as Data file).

**Funding:** This work was funded by the Natural Science Foundation of Shandong Province (Award Number: ZR2024MG049) to JW, Natural Science Foundation of Shandong Province (Award Number: ZR2021QG035) to JW, the Shandong Province Youth Entrepreneurship and Technology Support Program for Higher Education Institutions (Award Number: 2024KJH066) the recipient of the grant (Xue Zhang) is not listed in the authors, but JW is listed as one of the participants. The funders had a role in study design, data collection and analysis, decision to publish, or preparation of the manuscript.

**Competing interests:** The authors have declared that no competing interests exist.

## Conclusions

Based on the research results, this paper provides targeted suggestions for enterprises crisis response and their sustainable development under the new media environment.

## 1. Introduction

The rapid development of digital intelligence technology under the new media environment offers enterprises unprecedented opportunities, attracting more attention than ever before. Enterprises can utilize the benefits of the information age to promote products, enhance brand awareness, organize online events, and boost consumer engagement. This approach ensures comprehensive information coverage and enables rapid, precise delivery to stimulate consumption effectively. However, the new media environment has also facilitated the rapid spread of negative information about enterprises. The explosive *network public opinion information* (referred as *public opinion*) resulting from major crisis events can pose significant threats to their survival and development, especially in the current context where 5.5 billion netizens worldwide are highly active on social platforms.

Recent investigative reports by mainstream media have covered a wide range of industries, such as the Internet, automotive, finance, food, healthcare, home appliances, and digital sectors, exposing many crisis issues with well-known brands like BMW, Midea, Burger King, and Carrefour. These reports have revealed several problems, including product and service quality issues, inappropriate spokespersons, plagiarism and intellectual property violations, commercial competition disputes, and improper recruitment practices. The types of enterprises crisis show a decentralized feature, with various kinds of crisis emerging one after another. Moreover, different types of crises will trigger different degrees of netizens' attention and generate differentiated negative public opinion. However, enterprises often face limited resources for crisis management and public opinion guidance. Given these contradictions, it is crucial to explore how to effectively allocate crisis management resources to achieve optimal outcomes, reduce the intensity of negative public opinion, restore corporate image, and support sustainable business development.

## 2. Literature review

Public opinion arising from enterprises crisis events involves netizens as the subjects, the events as the object, and the social platforms as spreading carriers. Focusing on the above issues, through a systematic review of relevant literature and with the help of bibliometric analysis, it is found that the current academic research on enterprises crisis public opinion mainly focuses on the following three aspects.

**The generation mechanism of public opinion for enterprises crisis.** Based on the social exchange theory, the complex causal mechanism of the negative bias of enterprises crisis was discussed [1]. Similarly, the key elements driving surges in public opinion and their interaction paths were identified with the help actor-network

theory and qualitative comparative analysis strategies [2]. From the perspective of digital transformation, deep learning models were leveraged to develop a public opinion attention index, exploring its relationship with corporate digital transformation and green investment behavior [3]. The influence of positive social emotions on the digital transformation of manufacturing enterprises was empirically analyzed with a panel fixed-effects regression model [4]. Based on information ecology perspective, three channels through which public opinion affects small and medium-sized enterprises production safety were identified, that is, public awareness, media response, and government guidance [5].

**The evolutionary trends of crisis-derived public opinion.** a conceptual model of double-layer *online social networks* (OSN) leveraging the structural characteristics of OSN was developed to analyze the propagation rules of public opinion [6]. Similarly, the equilibrium conditions of stakeholder behavioral strategies were solved and detailed [7] and they provided management approaches and critical intervention points for public opinion communication. The opinion that social media reshaped traditional crisis research and contributed to secondary crisis propagation, influencing crisis management was proposed [8].

**The modelling of crisis-derived public opinion.** Based on their study of theoretical crisis management models, enhancing enterprises resilience and employing both reactive and forward-looking strategies significantly can improve crisis management performance [9]. From the perspective of complex networks, a competitive public opinion propagation model which integrating actual spreading scenarios was proposed and the key factors driving the synergistic evolution process were identified [10]. Based on game theory, a tripartite evolutionary game model including government, *social networking providers* (SNPs) and *we-media practitioners* (WPs) was constructed [11], and it was found that reasonable rewards and punishments can promote SNPs to participate in public opinion governance. Combing the SEIR (Susceptible, Exposed, Infected, Recovered) model and the evolutionary game model, the influence laws of public opinion spread on the tripartite evolutionary game was explored [12].

**The response to public opinion on enterprises crisis.** Based on the crisis management mode of big data, they demonstrated that big data technology enhances corporate data processing and introduced a new algorithm to predict crises in small and medium-sized enterprises [13]. Similarly, the crisis early warning in an online low-carbon tourism supply chain was explored supported by Bellman's continuous dynamic programming theory and big data marketing [14]. Considering the risk early warning, they suggested incorporating weak signals into their model for enterprise public opinion risk early warning, aiding enterprises in identifying potential public opinion risks [15]. From the perspective of trust repair, the effective strategies for repairing public trust, offering recommendations for enterprises to manage public opinion crises was explored through text mining and empirical testing [16]. Taking user characteristics as the entry point, the random forest algorithm and user profiling technology were innovatively applied to identify and target high-risk user groups likely to generate negative comments, thereby enhancing enterprise monitoring capabilities [17]. Similarly, to address public opinion challenges, the need for enterprises to establish comprehensive mechanisms for public opinion research, management, prevention, and recovery was systematically discussed from the e-business perspective [18–19]. Meanwhile, they experimentally found that celebrity tweets can restore public perception of enterprises, helping them manage crises and rebuild their image [20]. Based on the system dynamics, integrating the balanced scorecard and system dynamics, an enterprise public opinion performance management model was developed providing both a theoretical foundation and practical pathway for optimizing public opinion management strategies [21].

From the above literature review, it can be known that the rational utilization of resources for public opinion management to assist enterprises in responding to crisis is an important means of enterprises decision-making at present and the mainstream research paradigm in this field. However, it is found that there is still room for further research in this field. **(1)** Research on public opinion propagation and guidance based on game theory, primarily focused on the limited strategy choices among stakeholders, while few studies have examined how varying degrees of strategy choices affect the effectiveness of public opinion guidance. The differential game model, which captures the varying degrees of the same strategy among stakeholders offers a more accurate representation of public opinion dynamics. **(2)** Existing studies mainly

focused on the spreading characteristics and coping strategies for a single type of crisis, with limited attention to the differences between crisis types, and few studies consider the resources constraints of enterprises when facing public opinion crisis. Under the new media environment, enterprises crises are becoming increasingly diversified, and companies often face resources limitations in managing public opinion crisis. Therefore, how to allocate limited resources across different types of crises and maximize the effectiveness of governance resources is a critical issue in corporate management.

Based on the above issues, we classified the types of crises based on their content and introduced a differential game model involving enterprises, media and netizens in Section 3. Then, the strategic choices of various stakeholders under different decision-making scenarios are solved and discussed in Section 4. The evolution trends of each state variable and the allocation of enterprise governance crisis resources are analyzed through numerical simulation in Section 5. The research conclusions are summarized in Section 6.

## 3. Problem description and basic assumptions

### 3.1. Problem description

Under the context of the proliferation of self-media and social media, the public opinion landscape is complex and dynamic, with various types of crises, including financial, legal, credibility, and talent crises. The complexity and variety of enterprises public opinion crises intensify reputational damage and increase the complexity and urgency of the crisis management. These crises also significantly disrupt daily operations and long-term strategic development. Therefore, actively managing and reshaping the enterprises image, as well as quickly and effectively addressing public opinion crises, have become critical issues that enterprises must urgently address following the outbreak of crisis.

When enterprises addressed public opinion derived from crisis, a game system involving enterprises, online media (media) and netizens emerges and each subject will engage in games and updated their strategies. Media and netizens primarily serve as initiators and spreaders, generating and spreading relevant information to mitigate the crisis and engaging with the enterprises' response to the event. As the primary propagation medium for public opinion, media reports significantly influence public perception of key social events, which in turn shapes netizens' spreading choices and the evolution trend of the crisis. Netizens participate in discussions of trending events through online spectating, and their active engagement accelerates the spread of public opinion. In this system, enterprises act as operators, adjusting their guidance strategies for media and netizens to promote the reporting and propagation of truthful information. By collaborating in managing public opinion, they can steer the evolution of public opinion and resolve the crisis.

Focusing on the public opinion derived from enterprises crisis, this paper takes the real and positive public opinion (hereinafter referred to as public opinion) that is beneficial to enterprises as the research object, such as clarifying misunderstandings, addressing rumors, issuing apologies, rectifying mistakes, and similar initiatives. In practice, variations in the level of strategic effort under the same strategy can lead to different extents of public opinion propagation. Therefore, this study introduces the degree of strategic effort in the differential game to characterize the strategic choices of media and netizens. Specifically, strategic effort is represented by the reporting efforts of media and the propagation efforts of netizens. Additionally, the model's coefficient representing corporate subsidies to media and netizens serves as a crucial reference for enterprises in crisis management.

The public opinion spreading system consists of enterprises, media, and netizens. Enterprises crisis evolution and management are framed as strategic choices within a differential game model. Due to the diversification of social trends and the rapid development of social platforms, enterprises public opinion increasingly exhibits a typical risk amplification effect. Under this context, the likelihood of enterprises encountering crisis increases significantly, and these crises become increasingly complex. This paper focuses on how enterprises should adjust their strategic choices, categorize and understand the characteristics of different crisis events, and conduct targeted crisis planning with limited resources. Additionally, enterprises leverage the power of media and netizens to maximize rational and positive public opinion, enhance profitability, strengthen crisis management capabilities, and improve crisis management systems to navigate various crises effectively.

## 3.2. Crisis classification

In recent years, the crisis risks faced by enterprises have grown increasingly complex and diverse. A comprehensive review of crisis cases across industries shows that the types of crisis events have become more diversified, spanning a wide range of dimensions. This trend demands that enterprises be more responsive and adaptive in crisis management to effectively address various unexpected challenges. Combining the actual cases of enterprises crises outbreaks in recent years with the existing literature [22], this paper examines the following four common types of crises.

- **Product-type crises:** issues related to products and services. Currently, product and service issues remain the primary type of crisis faced by enterprises. Products and services remain the primary source of public opinion crises due to safety failures, functionality issues, counterfeiting, plagiarism, and other threats to consumer safety. The safety of products and services has become a fundamental baseline for crisis management. For example, the Guangzhou Toyota Camry, once a popular model in the B-class sedan market, has seen its sales drop by as much as 62% in recent years. Specifically, product quality issues, such as system failures, lagging performance, and inability to upgrade, have become the focus of consumer complaints.

- **Values-type crises:** issues related to enterprises political positions, cultural differences, conflicts, lack of respect, and other value-based concerns. The fate of enterprises is closely tied to national political security and mainstream ideology. A misaligned position or ambiguous stance on key events can easily provoke questions and criticisms online. Furthermore, if enterprises fail to respect or understand local culture, folklore, and values during product design, production, and marketing, they risk triggering a brand collapse. For example, on November 17, 2018, the Italian brand Dolce & Gabbana released a series of videos titled 'Start Chopsticks and Eat,' which were accused of racial discrimination, insulting Chinese culture, and misogyny. The designer's abusive remarks and the founder's arrogant apology further triggered widespread dissatisfaction among netizens. On the evening of November 21st, eight major e-commerce platforms in China, including Jingdong and Suning, removed all D&G products, and have yet to reinstate them. E-commerce sales were cut off due to a Chinese consumer boycott, causing the wealth of Dolce & Gabbana's two founders to shrink by 30%, and both fell off the 2018 Forbes billionaire list.

- **Internal management-type crises:** issues such as employees being treated unfairly, inappropriate remarks and behaviors by senior executives, and other internal management problems. In the digital age, inappropriate remarks and behaviors by founders and executives can amplify corporate risks. For instance, on May 31, 2024, Yu Minhong, the founder of New Oriental, commented on Oriental Selection and his personal career plan during a live broadcast, causing the company's share price to drop for five consecutive days, with a cumulative decline of 21.83% and a loss of over HK$4 billion in market value.

- **Marketing-type crises:** issues related to event operations, celebrity endorsements, and other marketing challenges. While activity planning and celebrity endorsements can drive traffic to a company, they also increase the risk of a persona collapse due to operational errors or the collapse of a spokesperson's personality and values, which can lead to a significant negative public opinion crisis. For instance, on September 10, 2023, during a live broadcast for Florasis, Li Jiaqi questioned netizens, claiming that eyebrow pencils were expensive because they did not work seriously, which led to stagnation in wages. Following this incident, sales at the Florasis online store dropped by more than 90%.

The above four types of enterprises crises are summarized as Table 1.

## 3.3. Basic assumptions

In order to describe the strategic choices of various stakeholders under different types of crises and represent the guiding effect of public opinion, the following variables are introduced in this sections.

**Table 1. The introduction of four types of enterprises crises.**

| Crisis type | Characteristics | Representative events | Frequencies* | Causes |
|---|---|---|---|---|
| Product-type | Directly related to consumer rights, a "disaster area" in crisis | Samsung Note7 explosion (2016); The BMW MINI Ice Cream Incident (2023) | 35%−40% | Direct contact with consumers makes quality issues or service loopholes more likely to be exposed by the media |
| Values-type | Society's expectations for enterprises' responsibility have risen | The Starbucks Racial Discrimination Incident (2018); The D&G Insulting China Incident (2018) | 15%−20% | The popularization of ESG concepts has drawn much attention to controversial events |
| Internal management-type | As enterprises expand in scale and supervision becomes stricter, the frequency of outbreaks increases | Luckin Coffee's financial fraud (2020); Internet Layoff Wave (2023) | 20%−25% | The expansion of an enterprise's scale does not match its management capabilities |
| Marketing-type | from an occasional occurrence to the norm. | Erke (2021); Durex's marketing failure (2018) | 25%−30% | Social media amplifies the spreading effect |

The data on the frequency of crisis occurrence is derived from statistical cases of enterprises crises in the past 10 years.

For enterprises, after crisis, they will provide relevant guidance and subsidy measures to encourage media and netizens to spread truthful and positive information to clarify the truth. The subsidy coefficient of enterprises to media and netizens under type $i$ crisis can be expressed and $U_{i1}$ and $U_{i2}$.

For media, what they need to decide is the degree of their efforts to report and spread positive public opinion on social platforms after the occurrence of crisis $i$, denoted by $E_{Mi}(t)$. During the reporting process of media, their costs are mainly reflected in aspects such as tracking and reporting on events and uncovering the truth, which is a convex function of their efforts, represented by $C_{Mi}(t) = \frac{\eta_M}{2}(1 - U_{i1})E_{Mi}^2(t)$. Among them, $\eta_M$ is the coefficient between the degree of media's effort and their cost. When enterprises didn't provide subsidy to media, their cost function becomes $C_{Mi}(t) = \frac{\eta_M}{2}E_{Mi}^2(t)$.

Similarly, for netizens, they need to decide the degree of their efforts to spreading positive public opinion in social platforms after crisis $i$, represented by $E_{Ni}(t)$. And during the spreading process of public opinion, the costs for netizens mainly relies in the time and energy they spend paying attention to, collecting, organizing and spreading, which is also a convex function of their efforts, denoted by $C_{Ni}(t) = \frac{\eta_N}{2}(1 - U_{i2})E_{Ni}^2(t)$. Among them, $\eta_N$ is the coefficient between the degree of netizens' effort and their cost. When enterprises didn't provide subsidy to the, their cost function becomes $C_{Ni}(t) = \frac{\eta_N}{2}E_{Ni}^2(t)$.

$R_M(t)$: Credibility of media. Netizens on social media platforms assess the degree of recognition of media according to its propagation power, influence, leading ability and other elements. For example, the higher the number of fans, the more recognized it is by netizens. Media guide positive public opinion and enhance credibility by actively reporting the truth. Based on the current mainstream research paradigm [23–25], It can be specifically expressed as:

$$\dot{R}_M(t) = \sum_{i=1}^{n} \varepsilon_M E_{Mi} - \mu R_M(t), \; \dot{R}_{M1}(t) = \sum_{i=1}^{n} (\varepsilon_M + \omega)E_{Mi} - \mu R_M(t)$$

(1)

In Eq. (1), $R_M(t)$ and $R_{M1}(t)$ represent the credibility of media in the unguided and explicitly guided, the dual-guided scenarios, respectively. $R_M(0) = R_{M0}$. $\varepsilon_M$ represents the influence coefficient of the degree of effort in reporting of media on its credibility; $\mu > 0$ means the impact of the public's forgetfulness or natural degradation, and $\omega$ is the implicit guidance coefficient.

$R_N(t)$: Netizen credibility. The degree of recognition of a netizen by other netizens in social platform based on comprehensive assessment of the netizens' behavior. They help enterprises deal with crises by actively spreading public opinion, thereby enhancing their credibility. It can be specifically expressed as:

$$\dot{R}_N(t) = \sum_{i=1}^{n} \varepsilon_N E_{Ni} - \mu R_N(t), \ \dot{R}_{N1}(t) = \sum_{i=1}^{n} (\varepsilon_N + \omega) E_{Ni} - \mu R_N(t)$$

(2)

In Eq. (2), $R_N(t)$ and $R_{N1}(t)$ represent the credibility of netizens in the unguided and explicitly guided, the dual-guided scenario. $R_N(0) = R_{N0}$. $\varepsilon_N$ represent the impact of the degree of effort exerted by netizens on their credibility, $\mu > 0$ means the impact of the public's forgetfulness or natural degradation.

$\pi_{Mi}$, $\pi_{Ni}$, $\pi_{Ei}$: The marginal revenue of media, netizens and enterprises under type $i$ crisis. The revenue of media mainly comes from aspects such as attracting netizens' attention, enhancing credibility and increasing traffic. The benefits for netizens mainly lie in enhancing their popularity. Enterprise profits are reflected in establishing a positive public influence, reducing the negative impact of crisis and expanding brand awareness.

$G_i(t)$: Traffic attention and heat accumulated of type $i$ crisis. It is a combination of multidimensional factors such as the number of posts, the frequency of views, the number of likes and so on. The level of attention is influenced by both media efforts and netizen engagement, undergoing a dynamic evolution that naturally declines over time. This relationship can be expressed as follows:

$$\dot{G}_i(t) = r_{Mi} E_{Mi} + r_{Ni} E_{Ni} - \delta_i G_i(t)$$

(3)

In Eq. (3), $r_{Ni}$ and $r_{Mi}$ denote the influence of netizens' and media's efforts on public opinion attention during type $i$ crisis, respectively, while $\delta_i > 0$ represents the effect of public forgetting or natural decline in attention following the crisis.

$Q_i(t)$: The spreading range of public opinion under type $i$ crisis, reflecting the effect of guiding public opinion, is commonly used as an indicator to assess the effectiveness of public opinion management. This paper introduces the variable of public opinion spreading range to accurately capture the interest of media and netizens during the propagation process, which can be expressed as follows:

$$Q_i(t) = a_i + \theta G_i(t) + \lambda_M R_M(t) + \lambda_N R_N(t)$$

(4)

In Eq. (4), $a_i$ denotes the basic spreading of type $i$ crisis, $\theta$ denotes the influence coefficient of public opinion concern on spreading range, $\lambda_M$ and $\lambda_N$ represent the influence coefficient of media credibility and netizen credibility on spreading range.

$p$: discount rate.

## 4. Game model construction and solution under different decision-making scenarios

When a crisis occurs, some media and netizens typically take the lead in releasing public opinion to compete for traffic dividends and maximize their own profit, creating a decentralized decision-making environment. However, when a core decision-making entity, such as enterprise, it may uniformly coordinate the behavior of media and netizens, aiming to quickly restore the facts and optimize the spreading scope and influence of public opinion, forming a centralized decision-making scenario.

To effectively guide public opinion after crisis, enterprises often employ strategies to encourage media and netizens to create and spread public opinion. Enterprises can provide detailed information to help media and netizens overcome challenges in verifying facts or offer subsidies to reduce their financial burden. This direct intervention and facilitation of public opinion propagation is termed the explicit guidance strategy. Based on the explicit guidance strategy, increasing the promotion of public opinion released by media and netizens with the help of intelligent technologies such as algorithmic recommendation not only enables media to gain traffic revenue, but also enhances public recognition and strengthens their credibility, which can be regarded as implicit guidance. This dual effect indirectly encourages their participation,

creating a virtuous cycle of disseminating public opinion and fostering positive interaction, referred to as the dual guidance strategy model.

According to the general principle of differential games [23], centralized decision-making model in an unconstrained environment typically yields higher system benefits than decentralized decision-making. However, considering the complex factors such as information asymmetry in real world, decentralized decision-making scenarios are more widely applied compared to the centralized decision-making scenarios in ideal states. To discuss the effectiveness of different guidance strategies, this paper introduced the decentralized decision-making scenario without guidance strategy as the benchmark. In addition, both the explicit guidance strategy and the dual guidance strategy are cost-sharing contractual situations, and they exhibit similar game dynamics when governed by the same contractual framework. Based on the above analysis, the study focuses on four public opinion crisis guidance strategies: decentralized decision-making without guidance strategy (*f*), decentralized decision-making with explicit guidance strategy (*xf*), centralized decision-making with explicit guidance strategy (*xj*), and decentralized decision-making with dual guidance strategy (*df*).

## 4.1. Decentralized decision-making without guidance strategy (*f*)

In this scenario, both media and netizens decided their degree of effort in reporting or spreading public opinion, aiming to maximize their interests. The Hamilton-Jacobi-Bellman (HJB) [24] equation was introduced to solve the following models. Assuming all parameters do not vary over time, and the explicit labeling of the time variable $t$ will be omitted in the following discussion [25]. Under this scenario, after crisis, the HJB equations for media and netizens are shown as follows:

$$pV_M^f = \max \sum_{i=1}^{n} \left\{ \pi_{Mi} \left[ a_i + \theta G_i^f(t) + \lambda_M R_M^f(t) \right] - \frac{\eta_M}{2} E_{Mi}^{f\,2}(t) + \frac{\partial V_M^f}{\partial G_i^f} \left[ r_{Mi} E_{Mi}^f + r_{Ni} E_{Ni}^f - \delta_i G_i^f(t) \right] + \right.$$
$$\left. \frac{\partial V_M^f}{\partial R_M^f} \left[ \sum_{i=1}^{n} \varepsilon_M E_{Mi}^f - \mu R_M^f(t) \right] \right\}$$

(5)

$$pV_N^f = \max \sum_{i=1}^{n} \left\{ \pi_{Ni} \left[ a_i + \theta G_i^f(t) + \lambda_N R_N^f(t) \right] - \frac{\eta_N}{2} E_{Ni}^{f\,2}(t) + \frac{\partial V_N^f}{\partial G_i^f} \left[ r_{Mi} E_{Mi}^f + r_{Ni} E_{Ni}^f - \delta_i G_i^f(t) \right] + \right.$$
$$\left. \frac{\partial V_N^f}{\partial R_N^f} \left[ \sum_{i=1}^{n} \varepsilon_N E_{Ni}^f - \mu R_N^f(t) \right] \right\}$$

**Proposition 1:** After crisis, the optimal decisions of each subject under the *f* scenario are as follows:

$$E_{Mi}^f = \frac{c_{1i} r_{Mi} + c_2 \varepsilon_M}{\eta_M}, E_{Ni}^f = \frac{b_{1i} r_{Ni} + b_2 \varepsilon_N}{\eta_N}$$

(6)

The optimal value functions for the revenue of media and netizens are as follows:

$$J_M^f = \int_0^{+\infty} e^{-pt} V_M^f dt = \int_0^{+\infty} e^{-pt} \left( \sum_{i=1}^{n} c_{1i} G_i^f + c_2 R_M^f + c_3 \right) dt$$

(7)

$$J_N^f = \int_0^{+\infty} e^{-pt} V_N^f dt = \int_0^{+\infty} e^{-pt} \left( \sum_{i=1}^{n} b_{1i} G_i^f + b_2 R_N^f + b_3 \right) dt$$

where

$$c_{1i} = \frac{\pi_{Mi}\theta}{P+\delta_i}; \quad c_2 = \frac{\sum_{i=1}^{n} \pi_{Mi}\lambda_M}{P+n\mu};$$

$$c_3 = \frac{1}{p} \sum_{i=1}^{n} \left[ a_i\pi_{Mi} - \frac{\eta_M}{2} E_{Mi}^{f}{}^2(t) + c_{1i}\left(r_{Mi}E_{Mi}^f + r_{Ni}E_{Ni}^f\right) + c_2 \sum_{i=1}^{n} \varepsilon_M E_{Mi}^f \right];$$

$$b_{1i} = \frac{\pi_{Ni}\theta}{P+\delta_i}; \quad b_2 = \frac{\sum_{i=1}^{n} \pi_{Ni}\lambda_N}{P+n\mu};$$

$$b_3 = \frac{1}{p} \sum_{i=1}^{n} \left[ a_i\pi_{Ni} - \frac{\eta_N}{2} E_{Ni}^{f}{}^2(t) + b_{1i}\left(r_{Mi}E_{Mi}^f + r_{Ni}E_{Ni}^f\right) + b_2 \sum_{i=1}^{n} \varepsilon_N E_{Ni}^f \right]$$

Proposition 1 presents the dynamic feedback equilibrium strategy for media and netizens under non-cooperative game. In this scenario, the optimal decision of media in responding to various crises is positively correlated with its marginal revenue ($\pi_{Mi}$), the influence coefficient of public opinion concern on spreading range ($\theta$), the influence coefficient of media credibility on public opinion spreading range ($\lambda_M$) and the influence coefficient of media effort on credibility ($\varepsilon_M$). The optimal decision of netizens in responding to crises is positively correlated with their marginal revenue ($\pi_{Ni}$), the enterprise's marginal revenue ($\pi_{Ei}$), the influence coefficient of public opinion concern on spreading range ($\theta$), the influence coefficient of netizens credibility on public opinion spreading range ($\lambda_N$), and the coefficient of netizen effort on credibility ($\varepsilon_N$). Their optimal strategies are negative correlated with the natural decay index ($\delta_i$), the public forgetting coefficient ($\mu$), and the discount rate ($p$). The above conclusions are in line with expectations.

### 4.2. Decentralized decision-making with explicit guidance strategy (xf)

During crisis, the cyberspace quickly becomes a forum for diverse public opinion. To effectively guide public opinion and enhance positive marketing penetration, enterprises, as system operators, may adapt explicit guidance strategy. This strategy seeks to mobilize media and netizens to promote public opinion supporting positive image under the enterprises' guidance. Enterprises then adjusts their decision-making to maximize overall benefits and achieve Pareto improvement in the performance of each participant. The decision-making interaction between enterprises, media, and netizens follows the Stackelberg game model.

The decision-making process is divided into two stages. First, enterprises need to determine the subsidy coefficients to media ($U_{i1}$) and netizens ($U_{i2}$) based on the type of crisis. Then, media and netizens need to decide their degree of effort in reporting and spreading public opinion based on their subsidy coefficients provided by enterprises. Therefore, the HJB equations for enterprises, media, and netizens in this scenario are as follows:

$$pV_E^{xf} = \max \sum_{i=1}^{n} \left\{ \pi_{Ei}\left[ a_i + \theta G_i^{xf}(t) + \lambda_M R_M^{xf}(t) + \lambda_N R_N^{xf}(t)\right] - \frac{\eta_M}{2} U_{i1}^{xf} E_{Mi}^{xf}{}^2(t) - \frac{\eta_N}{2} U_{i2}^{xf} E_{Ni}^{xf}{}^2(t) + \right.$$
$$\left. \frac{\partial V_E^{xf}}{\partial G_i^{xf}} \left[ r_{Mi}E_{Mi}^{xf} + r_{Ni}E_{Ni}^{xf} - \delta_i G_i^{xf}(t)\right] + \frac{\partial V_E^{xf}}{\partial R_M^{xf}} \left[ \sum_{i=1}^{n} \varepsilon_M E_{Mi}^{xf} - \mu R_M^{xf}(t)\right] + \frac{\partial V_E^{xf}}{\partial R_N^{xf}} \left[ \sum_{i=1}^{n} \varepsilon_N E_{Ni}^{xf} - \mu R_N^{xf}(t)\right] \right\}$$

$$pV_M^{xf} = \max \sum_{i=1}^{n} \left\{ \pi_{Mi}\left[ a_i + \theta G_i^{xf}(t) + \lambda_M R_M^{xf}(t)\right] - \frac{\eta_M}{2}\left(1-U_{i1}^{xf}\right) E_{Mi}^{xf}{}^2(t) + \frac{\partial V_M^{xf}}{\partial G_i^{xf}} \left[ r_{Mi}E_{Mi}^{xf} + r_{Ni}E_{Ni}^{xf} - \delta_i G_i^{xf}(t)\right] + \right.$$
$$\left. \frac{\partial V_M^{xf}}{\partial R_M^{xf}} \left[ \sum_{i=1}^{n} \varepsilon_M E_{Mi}^{xf} - \mu R_M^{xf}(t)\right] \right\}$$

(8)

$$pV_N^{xf} = \max \sum_{i=1}^{n} \left\{ \pi_{Ni}\left[ a_i + \theta G_i^{xf}(t) + \lambda_N R_N^{xf}(t)\right] - \frac{\eta_N}{2}\left(1-U_{i2}^{xf}\right) E_{Ni}^{xf}{}^2(t) + \frac{\partial V_N^{xf}}{\partial G_i^{xf}} \left[ r_{Mi}E_{Mi}^{xf} + r_{Ni}E_{Ni}^{xf} - \delta_i G_i^{xf}(t)\right] + \right.$$
$$\left. \frac{\partial V_N^{xf}}{\partial R_N^{xf}} \left[ \sum_{i=1}^{n} \varepsilon_N E_{Ni}^{xf} - \mu R_N^{xf}(t)\right] \right\}$$

**Proposition 2:** The optimal decisions of each subject under the **xf** scenario are as follows:

$$E_{Mi}^{xf} = \frac{h_{1i}r_{Mi} + nh_2\varepsilon_M}{\eta_M\left(1 - U_{i1}^{xf}\right)}; \quad E_{Ni}^{xf} = \frac{o_{1i}r_{Ni} + no_2\varepsilon_N}{\eta_N\left(1 - U_{i2}^{xf}\right)} \tag{9}$$

$$U_{i1}^{xf} = \frac{r_{Mi}\left(2j_{1i} - h_{1i}\right) + \varepsilon_M\left(2j_2 - h_2\right)}{r_{Mi}\left(2j_{1i} + h_{1i}\right) + \varepsilon_M\left(2j_2 + h_2\right)}; \quad U_{i2}^{xf} = \frac{r_{Ni}\left(2j_{1i} - o_{1i}\right) + \varepsilon_N\left(2j_3 - h_2\right)}{r_{Ni}\left(2j_{1i} + o_{1i}\right) + \varepsilon_N\left(2j_3 + h_2\right)}$$

The optimal value functions for the revenue of enterprises, media and netizens are as follows:

$$J_E^{xf} = \int_0^{+\infty} e^{-pt}V_E^{xf} = \int_0^{+\infty} e^{-pt}\left(\sum_{i=1}^n j_{1i}G_i^{xf} + j_2R_M^{xf} + j_3R_N^{xf} + j_4\right)dt$$

$$J_M^{xf} = \int_0^{+\infty} e^{-pt}V_M^{xf}dt = \int_0^{+\infty} e^{-pt}\left(\sum_{i=1}^n h_{1i}G_i^{xf} + h_2R_M^{xf} + h_3\right)dt \tag{10}$$

$$J_N^{xf} = \int_0^{+\infty} e^{-pt}V_N^{xf}dt = \int_0^{+\infty} e^{-pt}\left(\sum_{i=1}^n o_{1i}G_i^{xf} + o_2R_N^{xf} + o_3\right)dt$$

where

$$j_{1i} = \frac{\pi_{Ei}\theta}{P+\delta_i}; \quad j_2 = \frac{\sum_{i=1}^n \pi_{Ei}\lambda_M}{P+n\mu}; \quad j_3 = \frac{\sum_{i=1}^n \pi_{Ei}\lambda_N}{P+n\mu}$$

$$j_4 = \frac{1}{p}\sum_{i=1}^n\left[a_i\pi_{Ei} - \frac{\eta_M}{2}U_{i1}^{xf}E_{Mi}^{xf\,2}(t) - \frac{\eta_N}{2}U_{i2}^{xf}E_{Ni}^{xf\,2}(t) + j_{1i}\left(r_{Mi}E_{Mi}^{xf} + r_{Ni}E_{Ni}^{xf}\right) + j_2\sum_{i=1}^n\varepsilon_M E_{Mi}^{xf} + j_3\sum_{i=1}^n\varepsilon_N E_{Ni}^{xf}\right]$$

$$h_{1i} = \frac{\pi_{Mi}\theta}{P+\delta_i}; \quad h_2 = \frac{\sum_{i=1}^n \pi_{Mi}\lambda_M}{P+n\mu};$$

$$h_3 = \frac{1}{p}\sum_{i=1}^n\left[a_i\pi_{Mi} - \left(1 - U_{i1}^{xf}\right)\frac{\eta_M}{2}E_{Mi}^{xf\,2}(t) + h_{1i}\left(r_{Mi}E_{Mi}^{xf} + r_{Ni}E_{Ni}^{xf}\right) + h_2\sum_{i=1}^n\varepsilon_M E_{Mi}^{xf}\right];$$

$$o_{1i} = \frac{\pi_{Ni}\theta}{P+\delta_i}; \quad o_2 = \frac{\sum_{i=1}^n \pi_{Ni}\lambda_N}{P+n\mu};$$

$$o_3 = \frac{1}{p}\sum_{i=1}^n\left[a_i\pi_{Ni} - \left(1 - U_{i2}^{xf}\right)\frac{\eta_N}{2}E_{Ni}^{xf\,2}(t) + o_{1i}\left(r_{Mi}E_{Mi}^{xf} + r_{Ni}E_{Ni}^{xf}\right) + o_2\sum_{i=1}^n\varepsilon_N E_{Ni}^{xf}\right]$$

From the above theoretical results, it can be known that under various crises, the optimal decisions of media and netizens are directly proportional to variables such as their marginal revenue ($\pi_{Mi}, \pi_{Ni}$), the subsidy coefficient provided by enterprises ($U_{i1}, U_{i2}$), the influence coefficient of their effort on credibility ($\varepsilon_M, \varepsilon_N$), the influence of their efforts on public opinion attention ($r_M, r_N$) and the influence coefficient of their credibility on public opinion spreading range ($\lambda_M, \lambda_N$). And they are inversely change with the natural decay index of public opinion ($\delta_2$) and the forgetfulness coefficient ($\mu$). The above conclusions are also in line with expectations.

In addition, Proposition 2 shows that in **xf** scenario, the three stakeholders, especially media and netizens, exhibit differentiated behavioral choices under the explicit guidance of enterprises. The above conclusion precisely reflects that during the guiding process of public opinion derived from crisis, the strategic choices of all stakeholders are aimed at

maximizing their own interests. For example, enterprises mainly rely on media and netizens to report or forward information to guide the trend of public opinion, and they need to consider how to achieve the best guidance effect with the minimum cost. While for media or netizens, their focus is how to spread and release hot events or topics to enhance their influence and social recognition.

### 4.3. Centralized decision-making with explicit guidance strategy (*xj*)

To alleviate negative external effects of information asymmetry, enterprises can also integrate the power of media and netizens uniformly and achieve the best public opinion guidance effect by leveraging the overall power of the media-netizens system. In this decision-making scenario, there exists a core decision-maker within the game system, who maximizes the interests of the entire system by regulating the effects of all stakeholders, aiming to achieve the best guidance effect of public opinion. The decision-making of enterprises and the sub-game system comprising media and netizens, also follows the Stackelberg game framework. The game proceeds as follows: enterprises determine the subsidy coefficient $U_{i1}$ and $U_{i2}$ based on the crisis type. Then, media and netizens make decisions on their reporting and spreading efforts, aiming to achieve the optimal total return of the sub-game system. The HJB equations for enterprises and the sub-game system comprising media and netizens in this scenario are expressed as follows:

$$pV_E^{xj} = \max \sum_{i=1}^{n} \{\pi_{Ei}\left[a_i + \theta G_i^{xj}(t) + \lambda_M R_M^{xj}(t) + \lambda_N R_N^{xj}(t)\right] - \frac{\eta_M}{2} U_{i1}^{xj} E_{Mi}^{xj\,2}(t) - \frac{\eta_N}{2} U_{i2}^{xj} E_{Ni}^{xj\,2}(t) +$$

$$\frac{\partial V_E^{xj}}{\partial G_i^{xj}}\left[r_{Mi}E_{Mi}^{xj} + r_{Ni}E_{Ni}^{xj} - \delta_i G_i^{xj}(t)\right] + \frac{\partial V_E^{xj}}{\partial R_M^{xj}}\left[\sum_{i=1}^{n} \varepsilon_M E_{Mi}^{xj} - \mu R_M^{xj}(t)\right] + \frac{\partial V_E^{xj}}{\partial R_N^{xj}}\left[\sum_{i=1}^{n} \varepsilon_N E_{Ni}^{xj} - \mu R_N^{xj}(t)\right]\}$$

(11)

$$pV_{MN}^{xj} = \max \sum_{i=1}^{n} \{(\pi_{Mi} + \pi_{Ni})\left[a_i + \theta G_i^{xj}(t) + \lambda_M R_M^{xj}(t) + \lambda_N R_N^{xj}(t)\right] - \frac{\eta_M}{2}(1 - U_{i1}^{xj})E_{Mi}^{xj\,2}(t) -$$

$$\frac{\eta_N}{2}(1 - U_{i2}^{xj})E_{Ni}^{xj\,2}(t) + \frac{\partial V_E^{xj}}{\partial G_i^{xj}}\left[r_{Mi}E_{Mi}^{xj} + r_{Ni}E_{Ni}^{xj} - \delta_i G_i^{xj}(t)\right] + \frac{\partial V_E^{xj}}{\partial R_M^{xj}}\left[\sum_{i=1}^{n} \varepsilon_M E_{Mi}^{xj} - \mu R_M^{xj}(t)\right] +$$

$$\frac{\partial V_E^{xj2}}{\partial R_N^{xj}}\left[\sum_{i=1}^{n} \varepsilon_N E_{Ni}^{xj} - \mu R_N^{xj}(t)\right]\}$$

**Proposition 3:** The optimal decisions of each subject in the *xj* scenario are as follows:

$$E_{Mi}^{xj} = \frac{v_{1i}r_{Mi} + nv_2\varepsilon_M}{\eta_M\left(1 - U_{i1}^{xj}\right)}, \; E_{Ni}^{xj} = \frac{v_{1i}r_{Ni} + nv_3\varepsilon_N}{\eta_N\left(1 - U_{i2}^{xj}\right)}$$

(12)

$$U_{i1}^{xj} = \frac{r_{Mi}(2u_{1i} - v_{1i}) + \varepsilon_M(2u_2 - v_2)}{r_{Mi}(2u_{1i} + v_{1i}) + \varepsilon_M(2u_2 + v_2)}, \; U_{i2}^{xj} = \frac{r_{Ni}(2u_{1i} - v_{1i}) + \varepsilon_N(2u_3 - v_3)}{r_{Ni}(2u_{1i} + v_{1i}) + \varepsilon_N(2u_3 + v_3)}$$

The optimal value functions for the revenue of enterprises and sub-game system consisting of media and netizens are as follows:

$$J_E^{xj} = \int_0^{+\infty} e^{-pt}V_E^{xj}dt = \int_0^{+\infty} e^{-pt}\left(\sum_{i=1}^{n} u_{1i}G_i^{xj} + u_2 R_M^{xj} + u_3 R_N^{xj} + u_4\right)dt$$

(13)

$$J_{MN}^{xj} = \int_0^{+\infty} e^{-pt} V_{MN}^{xj} dt = \int_0^{+\infty} e^{-pt} \left( \sum_{i=1}^n v_{1i} G_i^{xj} + v_2 R_M^{xj} + v_3 R_N^{xj} + v_4 \right) dt$$

where

$$u_{1i} = \frac{\pi_{Ei}\theta}{P+\delta_i}; \ u_2 = \frac{\sum_{i=1}^n \pi_{Ei}\lambda_M}{p+n\mu}; \ u_3 = \frac{\sum_{i=1}^n \pi_{Ei}\lambda_N}{p+n\mu},$$

$$u_4 = \frac{1}{p} \sum_{i=1}^n \left[ a_i\pi_{Ei} - \frac{\eta_M}{2} U_{i1}^{xj} E_{Mi}^{xj\,2}(t) - \frac{\eta_N}{2} U_{i2}^{xj} E_{Ni}^{xj\,2}(t) + u_{1i}\left( r_{Mi}E_{Mi}^{xj} + r_{Ni}E_{Ni}^{xj} \right) + u_2 \sum_{i=1}^n \varepsilon_M E_{Mi}^{xj} + u_3 \sum_{i=1}^n \varepsilon_N E_{Ni}^{xj} \right]$$

$$v_{1i} = \frac{(\pi_{Mi}+\pi_{Ni})\theta}{p+\delta_i}; \ v_2 = \frac{\sum_{i=1}^n (\pi_{Mi}+\pi_{Ni})\lambda_M}{p+n\mu}; \ v_3 = \frac{\sum_{i=1}^n (\pi_{Mi}+\pi_{Ni})\lambda_N}{p+n\mu};$$

$$v_4 = \frac{1}{p} \sum_{i=1}^n [a_i(\pi_{Mi}+\pi_{Ni}) - \frac{\eta_M}{2}(1-U_{i1}^{xj})E_{Mi}^{xj\,2}(t) - \frac{\eta_N}{2}(1-U_{i1}^{xj})E_{Ni}^{xj\,2}(t) +$$

$$v_{1i}\left( r_{Mi}E_{Mi}^{xj} + r_{Ni}E_{Ni}^{xj} \right) + v_2 \sum_{i=1}^n \varepsilon_M E_{Mi}^{xj} + v_3 \sum_{i=1}^n \varepsilon_N E_{Ni}^{xj}]$$

The inference result of Proposition 3 is basically consistent with the above text and will not be elaborated here.

### 4.4. Decentralized decision-making with dual guidance strategy (*df*)

In addition to explicit guidance, enterprises can enhance the visibility of media reports and public opinion released by netizens with the help of personalized algorithms. While obtaining traffic revenue, media and netizens can also strengthen their social recognition, that is, increasing the influence of the degree of their effort in reporting and spreading on its credibility (denoted as $\omega$), thus forming dual guidance scenario. From the above definition, in *df* scenario, the credibility functions of media and netizens need to be adjusted respectively, as shown in Eq. (1–2). In this context, enterprises' dual-guidance strategy fully leverages the positive role of media and netizens in guiding public opinion. The decision-making process is divided into two stages: first, enterprises determines the subsidy coefficients ($U_{i1}, U_{i2}$) based on the crisis type; media and netizens decide the degree of reporting and spreading efforts based on the given subsidy coefficients, aiming to maximize their own revenue.

Therefore, the HJB equations for enterprises, media, and netizens are as follows:

$$pV_E^{df} = \max \sum_{i=1}^n \{\pi_{Ei}\left[ a_i + \theta G_i^{df}(t) + \lambda_M R_M^{df}(t) + \lambda_N R_N^{df}(t) \right] - \frac{\eta_M}{2} U_{i1}^{df} E_{Mi}^{df\,2}(t) - \frac{\eta_N}{2} U_{i2}^{df} E_{Ni}^{df\,2}(t) +$$

$$\frac{\partial V_E^{df}}{\partial G_i^{df}}\left[ r_{Mi}E_{Mi}^{df} + r_{Ni}E_{Ni}^{df} - \delta_i G_E^{df}(t) \right] + \frac{\partial V_E^{df}}{\partial R_M^{df}}\left[ \sum_{i=1}^n \varepsilon_M E_{Mi}^{df} - \mu R_M^{df}(t) \right] + \frac{\partial V_E^{df}}{\partial R_N^{df}}\left[ \sum_{i=1}^n \varepsilon_N E_{Ni}^{df} - \mu R_N^{df}(t) \right]\}$$

$$pV_M^{df} = \max \sum_{i=1}^n \{\pi_{Mi}\left[ a_i + \theta G_i^{df}(t) + \lambda_M R_M^{df}(t) \right] - \frac{\eta_M}{2}(1-U_{i1}^{df}) E_{Mi}^{df\,2}(t) + \frac{\partial V_M^{df}}{\partial G_i^{df}}\left[ r_{Mi}E_{Mi}^{df} + r_{Ni}E_{Ni}^{df} - \delta_i G_i^{df}(t) \right] +$$

$$\frac{\partial V_M^{df}}{\partial R_M^{df}}\left[ \sum_{i=1}^n \varepsilon_M E_{Mi}^{df} - \mu R_M^{df}(t) \right]\} \tag{14}$$

$$pV_N^{df} = \max \sum_{i=1}^n \{\pi_{Ni}\left[ a_i + \theta G_i^{df}(t) + \lambda_N R_N^{df}(t) \right] - \frac{\eta_N}{2}(1-U_{i2}^{df}) E_{Ni}^{df\,2}(t) + \frac{\partial V_N^{df}}{\partial G_i^{df}}\left[ r_{Mi}E_{Mi}^{df} + r_{Ni}E_{Ni}^{df} - \delta_i G_i^{df}(t) \right] +$$

$$\frac{\partial V_N^{df}}{\partial R_N^{df}}\left[ \sum_{i=1}^n \varepsilon_N E_{Ni}^{df} - \mu R_{N1}^{df}(t) \right]\}$$

**Proposition 4:** The optimal decisions of each subject in **df** scenario are as follows:

$$E_{Mi}^{df} = \frac{r_{1i}r_{Mi} + r_2\left(\varepsilon_M + \omega\right)}{\eta_M\left(1 - U_{i1}^{df}\right)}, \; E_{Ni}^{df} = \frac{s_{1i}r_{Ni} + s_2\left(\varepsilon_N + \omega\right)}{\eta_N\left(1 - U_{i2}^{df}\right)}$$

(15)

$$U_{i1}^{df} = \frac{r_{Mi}\left(2q_{1i} - r_{1i}\right) + \left(\varepsilon_M + \omega\right)\left(2q_2 - r_2\right)}{r_{Mi}\left(2q_{1i} + r_{1i}\right) + \left(\varepsilon_M + \omega\right)\left(2q_2 + r_2\right)}, \; U_{i2}^{df} = \frac{r_{Ni}\left(2q_{1i} - s_{1i}\right) + \left(\varepsilon_N + \omega\right)\left(2q_3 - s_2\right)}{r_{Ni}\left(2q_{1i} + s_{1i}\right) + \left(\varepsilon_N + \omega\right)\left(2q_3 + s_2\right)}$$

The optimal value functions for the revenue of enterprises, media and netizens are as follows:

$$J_E^{df} = \int_0^{+\infty} e^{-pt} V_E^{df} dt = \int_0^{+\infty} e^{-pt}\left(\sum_{i=1}^n q_{1i}G_i^{df} + q_2 R_M^{df} + q_3 R_N^{df} + q_4\right) dt;$$

$$J_M^{df} = \int_0^{+\infty} e^{-pt} V_M^{df} dt = \int_0^{+\infty} e^{-pt}\left(\sum_{i=1}^n r_{1i}G_i^{df} + r_2 R_M^{df} + r_3\right) dt$$

(16)

$$J_N^{df} = \int_0^{+\infty} e^{-pt} V_N^{df} dt = \int_0^{+\infty} e^{-pt}\left(\sum_{i=1}^n s_{1i}G_i^{df} + s_2 R_M^{df} + s_3\right) dt$$

where

$$q_{1i} = \frac{\pi_{Ei}\theta}{P+\delta_i}; \; q_2 = \frac{\sum_{i=1}^n \pi_{Ei}\lambda_M}{P+n\mu}; \; q_3 = \frac{\sum_{i=1}^n \pi_{Ei}\lambda_N}{P+n\mu};$$

$$q_4 = \frac{1}{P}\sum_{i=1}^n\left[a_i\pi_{Ei} - \frac{\eta_M}{2}U_{i1}^{df}E_{Mi}^{df\,2}(t) - \frac{\eta_N}{2}U_{i2}^{df}E_{Ni}^{df\,2}(t) + q_{1i}\left(r_{Mi}E_{Mi}^{df} + r_{Ni}E_{Ni}^{df}\right) + q_2\sum_{i=1}^n\left(\varepsilon_M + \omega\right)E_{Mi}^{df}\right]$$
$$+q_3\sum_{i=1}^n\left(\varepsilon_N + \omega\right)E_{Ni}^{df}\right]$$

$$r_{1i} = \frac{\pi_{Mi}\theta}{P+\delta_i}; \; r_2 = \frac{\sum_{i=1}^n \pi_{Mi}\lambda_M}{P+n\mu}$$

$$r_3 = \frac{1}{P}\sum_{i=1}^n\left[a_i\pi_{Mi} - \left(1 - U_{i1}^{df}\right)\frac{\eta_M}{2}E_{Mi}^{df\,2}(t) + r_{1i}\left(r_{Mi}E_{Mi}^{df} + r_{Ni}E_{Ni}^{df}\right) + r_2\sum_{i=1}^n\left(\varepsilon_M + \omega\right)E_{Mi}^{df}\right]$$

$$s_{1i} = \frac{\pi_{Ni}\theta}{P+\delta_i}; \; s_2 = \frac{\sum_{i=1}^n \pi_{Ni}\lambda_N}{P+n\mu}$$

$$s_3 = \frac{1}{P}\sum_{i=1}^n\left[a_i\pi_{Ni} - \left(1 - U_{i2}^{df}\right)\frac{\eta_M}{2}E_{Ni}^{df\,2}(t) + s_{1i}\left(r_{Mi}E_{Mi}^{df} + r_{Ni}E_{Ni}^{df}\right) + s_2\sum_{i=1}^n\left(\varepsilon_N + \omega\right)E_{Ni}^{df}\right]$$

As shown in Proposition 4, in addition to the similar inferences mentioned above, the optimal decision of media and netizens also change positively with the implicit guidance coefficient ($\omega$). In fact, this decision-making scenario is closer to the practice of guiding and managing public opinion derived from crisis. Each stakeholder makes decisions with the aim of maximizing their own benefits. While enterprises reduce the reporting or spreading costs for media and netizens, they will

also increase the exposure rate of related public opinion through technical means such as algorithmic recommendations, thereby enhancing the attention paid to public opinion and assisting enterprises in crisis management.

### 4.5. Comparisons and analyses

To analyze the guiding process of public opinion derived from enterprises crisis under different decision-making scenarios, this section compares the optimal decisions of media and netizens, as well as enterprises subsidies and other parameters in the above scenarios, and reaches the following conclusions.

(1) In the scenario without guidance, media and netizens have the lowest degree of effort, and with the intervention of enterprises guidance strategies, their efforts have improved. Specifically, in **f** scenario, media and netizens each pursue their interest's maximization without collaboration and cost support, resulting in insufficient efforts and thus limiting spreading scope of public opinion. While in **xf** and **xj** scenarios, with the introduction of enterprises' subsidy coefficient, the reporting and spreading cost of public opinion for media and netizens has decreased, especially, in **xj** scenario, the cooperation between media and netizens has further reduced their costs, leading to an increase in their efforts. In **df** scenario, the cost subsidies and information diversion provided by enterprises not only reduce the public opinion spreading costs for media and netizens, but also enhance their traffic revenue and social recognition. Therefore, the effects of media and netizens have also improved.

(2) Media and netizens have higher degree of effort in centralized decision-making scenario, while enterprises need to pay higher subsidy coefficient in decentralized decision-making scenario. As for the reason, in decentralized decision-making scenario, media and netizens "fight on their own", pursuing the best benefits for themselves, and fail to collaborate. This leads to both stakeholders paying a relatively high cos for the propagation of public opinion, which in turn results in a lower degree of effort and a poor effect in guiding public opinion. In centralized decision-making scenario, for the sub-game system formed by media and netizens, in addition to enjoying subsidies provided by enterprises, their cooperation further reduces the cost of public opinion propagation. While expanding the spreading scope of public opinion, the improvement of their credibility also prompts them to actively participate in the cooperation process and have more effort. Besides, the subsidy for media and netizens provided by enterprises is relatively low, i.e., enterprises can obtain better public opinion guidance effects with a lower cost. This is also since the cooperation of the media and netizens reduces the cost of public opinion reporting (spreading), further stimulating the enthusiasm of both stakeholders.

(3) In centralized decision-making scenario, the credibility of media (netizens), the attention and spreading range of public opinion are all higher than those in decentralized decision-making scenario. The cooperation between media and netizens in centralized decision-making scenario, coupled with the enjoyment of enterprises subsidy, further reduces the cost of public opinion propagation and enhances their credibility, providing impetus for their reporting (spreading) public opinion and forming a virtuous cycle. This significantly expands the spreading scope of public opinion and reinforces the enterprises image.

(4) The implicit guidance coefficient ($\omega$) plays a key role when comparing the strategic choices of media and netizens in **xj** and **df** scenarios. When $\omega \geq \frac{1}{\sum\limits_{i=1}^{n} \pi_{Mi}} \left( \frac{r_{Mi}\pi_{Ni}\theta(P+n\mu)}{\lambda_M} + \varepsilon_M \sum\limits_{i=1}^{n} \pi_{Ni} \right)$, there exits $E_{Mi}^{xj} \leq E_{Mi}^{df}$ and vice versa.

When $\omega \geq \frac{1}{\sum\limits_{i=1}^{n} \pi_{Ni}} \left( \frac{r_{Ni}\pi_{Mi}\theta(P+n\mu)}{\lambda_N} + \varepsilon_N \sum\limits_{i=1}^{n} \pi_{Mi} \right)$, there exits $E_{Ni}^{xj} \leq E_{Ni}^{df}$ and vice versa. From the above inference, it can be known that the optimization of guiding effect of public opinion can be achieved by adjusting the cooperation form between media and netizens (i.e., **xj**), or by updating the incentive strategy (i.e., **df**). The main difference in their effect

lies in the extent to which implicit incentives affect their credibility. When $\omega$ is relatively large, that is, enterprises can significantly enhance the credibility of media and netizens through personalized recommendations, algorithmic traffic diversion and other technical means, thereby bringing them higher social recognition. At this point, compared with the reduction in spreading costs brought about by the cooperation between media and netizens in centralized decision-making, they are more willing to pursue high exposure and promotion rates of information, and thus will make greater efforts. Conversely, when $\omega$ is smaller, media and netizens are more willing to pursue a reduction in spreading costs.

Besides, it can be seen from the above results that the strategy choices and benefits of each stakeholders involve many factors under different decision-making scenarios, making theoretical analysis rather difficult. Numerical simulation will be conducted in the next chapter.

## 5. Numerical analysis

To illustrate the propositions and inferences more intuitively and compare the impacts of different strategies adopted by enterprises during crisis on the guiding effect of public opinion, numerical simulation was introduced to achieve the following three purposes. First, we will simulate the evolution trends of each variable over time in different scenarios to verify the above propositions and inferences. The second is to explore the best resources allocation plans for enterprises in response to different types of crises. The third is to discuss the impact of the subsidy coefficient of media and netizens on the guiding effect of public opinion, and to assist enterprises in making decisions.

To ensure the accuracy of the experimental results, we introduced the following three steps for setting the parameters. Firstly, based on research results [26,27] and actual data, the parameters were assigned as reference for subsequent steps. Then, a total of 17 experts, including professors, associate professors, and doctors in the field of public opinion and emergence management, were selected to investigate the parameters of the initial step. These experts can choose to agree with the assignment or provide their own opinions and update the results. Finally, a symposium involving 9 experts was held to fully discuss the rationality of the updated parameters until more than half of the experts approved, and the final assignment was determined. The main dimensionless parameters were set as follows:

$$\eta_M = 15, \ \eta_N = 10, \ p = 0.9, \ \mu = 0.9, \varepsilon_M = 2, \ \varepsilon_N = 1.5,$$
$$\lambda_M = 0.9, \ \lambda_N = 0.7, \ R_{M0} = 0.5, \ R_{N0} = 0.3, \ \omega = 1.9, \ \theta = 3 \tag{17}$$

### 5.1. Evolution trends of the variables in different scenarios

Based on the parameter defined in Eq. (17), the experimental results under different scenarios are shown in Fig 1.

Fig 1 shows the evolution process of various variables, such as the attention ($G$) and spreading range of public opinion ($Q$), the credibility of media ($R_M$) and netizens ($R_N$) and the benefits of each stakeholders ($V$) under four decision-making scenarios. Overall, all variables show similar evolutionary trend in the four scenarios. They increase slowly, then tend to stabilize and reach stable equilibrium over time.

Through comparing the evolutionary trends and results of the relevant variables in the four decision-making scenarios, we can get the following conclusions. The guidance effect of public opinion derived from crisis is significantly better in explicit and dual-guidance decision-making scenarios than that in unguided scenario, and the guiding effect reaches Pareto state in dual-guidance scenario.

Compared with other scenarios, the attention and spreading range of public opinion are highest in dual-guidance decision-making scenario, as shown in Fig 1(a) and 1(d). And at this point, the implicit guidance coefficient ($\omega$) is relatively high, which also verifies the fourth inference mentioned above. Furthermore, as shown in Fig 1(b)–1(c)) and 1(e)–1(f), when enterprises experience crisis, adopting the dual-guidance strategy not only effectively improves the credibility of media and netizens but also significantly increases the benefits of each stakeholder.

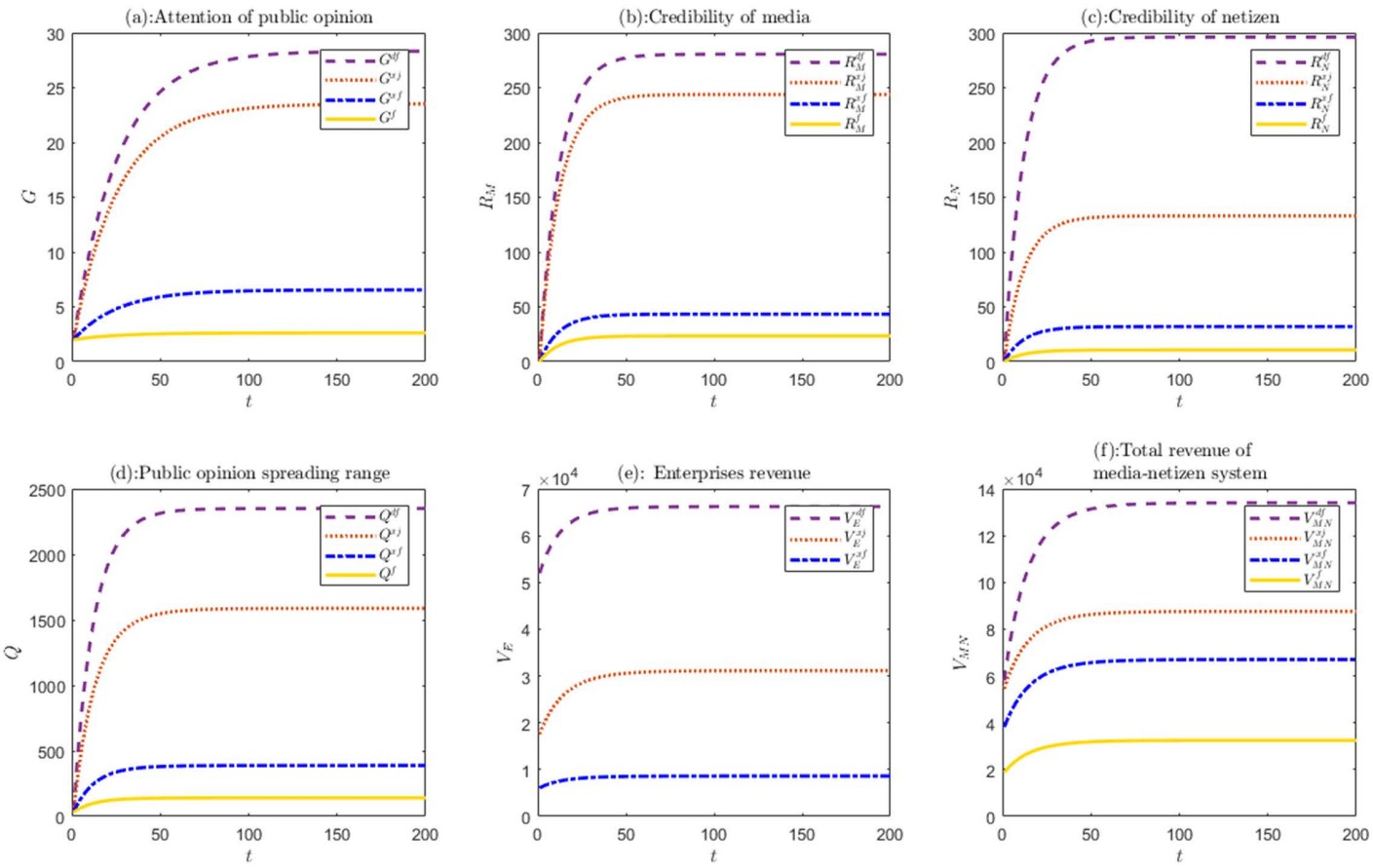

**Fig 1. The evolution trends of various variables under four scenarios.**

Besides, considering the information asymmetry in practical management, decentralized decision-making is more prevalent than the idealized centralized decision-making. And the centralized decision-making scenario consistently adopted in the whole crisis management process incurs high costs and is rather difficult to implement. Therefore, it is necessary to seek for higher benefits in decentralized decision-making. Through theoretical analysis and simulation experiments, as shown in Fig 1, the benefits for each subject in decentralized decision-making scenario with dual-guidance are significantly greater than those in centralized decision-making scenario with explicit guidance. This scenario can effectively enhance public opinion guidance and respond to public opinion crises more efficiently.

The above experimental results offer insights and guidance for optimizing enterprises strategies in managing public opinion crisis. Enterprises, as key public opinion influencers, should assess the situation carefully and make informed decisions post-crisis. On the one hand, they should fully mobilize the enthusiasm of media and netizens to participate in the spreading process of public opinion through cost subsidies and other means, taking the initiative in public opinion guidance and control, and improving the collaborative governance system of media and netizens. On the other hand, enterprises also need to rely on technologies such as algorithmic recommendation to further expand the spreading scope of public opinion and help them get through crisis. For example, Enterprises can adopt the dual-guidance strategy, leveraging the influence of media and netizens to broaden the reach of public opinion propagation. By taking proactive guidance of public opinion, enterprises can enhance the collaborative governance system between media and netizens, helping to overcome the public opinion crisis.

## 5.2. Evolution trends of variables in *df* scenario considering crisis differentiation

As discussed in Section 5.1, the guiding effect of public opinion is more effective in *df* scenario. Therefore, this section will discuss the evolution trends of various variables across different crises in *df* scenario, helping enterprises respond crisis actively and effectively. In this section, based on Eq. (17), $n = 4$ is introduced to represent the four types of crises and to discuss the priority responding order. In the previous definition, the main parameters affected by different types of crises are the marginal benefit of three stakeholders ($\pi$), the basic spreading range ($a$) and the forgetting rate of public opinion ($\delta$). Combining the characteristics of the four types of crisis, the relationship of above three parameters defined as follows.

$$\pi_{E2} > \pi_{E1} > \pi_{E3} > \pi_{E4},\ \pi_{M2} > \pi_{M4} > \pi_{M1} > \pi_{M3},\ \pi_{N4} > \pi_{N2} > \pi_{N1} > \pi_{N3},\ a_2 > a_4 > a_1 > a_3,\ \delta_4 > \delta_3 > \delta_1 > \delta_2 \quad (18)$$

Specifically, when $i = 1$, it represents product-type crises, which is closely related to enterprise. The relevant audience is primarily stakeholders, resulting in relatively low basic propagation range. The optimal response to this crisis is improving products and services, leading the efforts of media and netizens have a relatively low impact on public opinion attention. When $i = 2$, it represents values-type crises. Values are the foundation for enterprises' existence, and such crises have significant impacts on enterprise. Issues such as political stance and enterprises values affect a large of audience, leading to wider spreading range. This crisis type significantly impacts enterprises, and the publicity by media and netizens helps enhance their social recognition. Simultaneously, enterprises, media, and netizens can achieve high marginal returns. When $i = 3$, it represents internal management-type crises. These crises resonate with the public, and netizens' efforts significantly influence public opinion. The marginal benefit for media is lower, as their reporting of internal management crises may lead to public skepticism. When $i = 4$, it represents marketing-type crises. Media and netizens' publicity of these crises helps gain attention, resulting in higher marginal benefits for them. Celebrity-related events attract large fan bases increasing the basic spreading range. Due to the rapid turnover of celebrities, the public's tendency to forget this type of crisis is relatively high.

Based on above analysis, numerical simulations are carried out for four types of crises and the results are shown in Fig 2.

Fig 2 presents the evolution trends of key variables, such as the attention and spreading range of public opinion, the benefits of each subjects, in *df* scenario considering crisis differentiation. Overall, the variables of the four types of crises have similar evolutionary trends. Over time, they gradually increase, then tends to level off and eventually reach stable equilibrium state. Through comparing the evolution trends of variables of different types of crises, it can be concluded that the characteristics of public opinion derived from different types of crises vary, and they also have different impacts on enterprises.

As shown in Fig 2, in *df* decision-making scenario, the evolution trends of different types of crises follow similar pattern, with the values-type crises receiving the most attention and having the widest spreading range. When this kind of crisis occurs, such as political stance, it typically attracts widespread national attention, and even minor actions by the enterprise are amplified by media and netizens. Marketing-type and internal management-type crises also attract higher attention and result in broader spreading range of public opinion. The unique attributes of celebrities lead to greater attention to enterprise-related crises, and their large fan base facilitates the widespread propagation of positive public opinion about these crises. In recent years, as the public has become more focused on defending their rights, they are more likely to empathize with such crises and closely follow their development. Overall, product-type crises are more common for enterprises, with attention primarily focused on the stakeholders involved. Compared to other types of crises, public opinion regarding these events tends to be less widespread.

For enterprises, in *df* scenario, managing values-based crises yields the greatest benefits. Such crises attract significant public attention; timely and effective management not only reinforces positive values but also helps rebuild enterprises' image and enhances its public reputation. The subsequent type is "product-type" crises. A timely response from the enterprises to address issues, clarify misunderstandings, take responsibility, and implement corrective actions can help

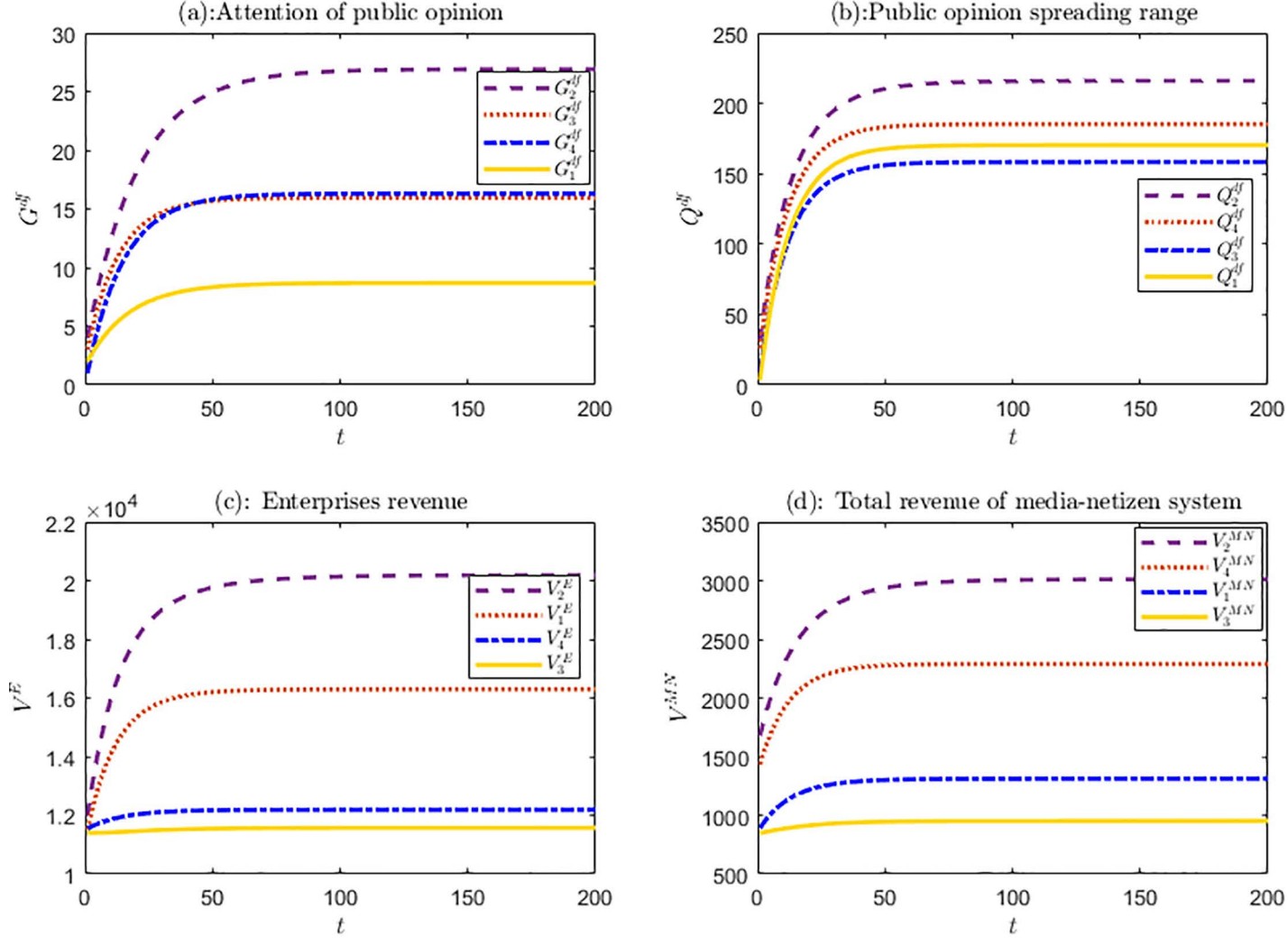

**Fig 2. The evolution trend of four types of crises in the *df* scenario.**

navigate this kind of crisis, restore public trust, and in some cases, serve as indirect publicity for its products or services. In contrast, managing marketing-type and internal management-type crises generates relatively low revenue. Issues related to celebrity endorsements often involve contractual disputes, irrational fan behavior and other complications, which hinder effective crisis management. Addressing internal management issues is time-consuming, costly, and uncertain, making the management of these crises both challenging and costly with limited benefits.

While for media and netizens, the priority order of media and netizens is the same as that of enterprises. As for the reason, values-type crises attract significant attention from media and netizens, increasing viewership. Social platforms prioritize promoting these crises to boost exposure and generate higher revenue. For the marketing-type crises, celebrity involvement attracts substantial traffic and a large of fans, drawing the attention of media and netizens when promoting crisis-related public opinion. Consequently, media and netizens, guided by enterprises, give more attention to this type of crisis, resulting in higher profits.

The experimental results offer insights and guidance for optimizing enterprises strategies in managing public opinion crisis. Facing different types of crises and limited governance resources, enterprises should allocate resources in a

targeted manner, give priority to handing value-type and product-type crises, and leverage media and netizens to amplify positive public opinion, preserve the company's image, and mitigate the negative effects.

## 5.3. Allocation proportion of various types of crises guided in *df* scenarios

The experimental results and analyses in Section 5.2 have helped enterprises determine the priority management order for guiding public opinion, i.e., enterprises should prioritize the values-type crises, followed by the product-type crises, marketing-type crises, and internal management-type crises. To determine the proportion of resources allocation for guiding four types of crises under the limited resources, we assumed that the handling processes of different types of crises are relatively independent, that is, different types of crises do not interfere with or influence each other during the handling process.

$g_i$ was introduced to represents the proportion of resources allocation for crisis $i$ and the total revenue of enterprise as the observed variable. With the help of numerical simulation, the priority is to determine the resources allocation ratio $g_2$ for managing this kind of crisis within the enterprises limited resources. Allocation ratios $g_1$, $g_4$ and $g_3$ for the remaining kinds pf crises are then set based on their priority. The experiment results are shown in Fig 3.

As shown in Fig 3, when the ratios of enterprises resources invested in values-type crises, product-type crises, marketing-type crises, and internal management-type crises are 60%, 28%, 8.4%, and 3.6% respectively, enterprises achieve optimal overall benefit. The experimental results are consistent with the above analysis. The proportion of resources for responding to values-type and product-type crises is as high as 88%, while the cumulative proportion of resources for responding to the other two types of crises is 12%.

In practice, enterprises should prioritize crisis management resources for value-type crises, leveraging media and netizen reports to spread information that promotes positive enterprises values, rebuild company's image, and ultimately improve profitability. Since high-quality products or services are the primary sources of revenue, enterprises should also allocate part of their management resources to addressing such kinds of crises. This will effectively promote the quality of their products, services, and other areas of public relations, thereby boosting corporate revenue.

## 5.4. The influence of subsidy coefficient on the guiding effect of public opinion

To further discuss the influence of the subsidy coefficients of enterprises to media and netizens on the guiding effect of public opinion derived from crisis, taking *df* scenario as example, we adjusted the changes in two types of subsidy coefficients, selected the spreading range of public opinion as the observation indicator. The experimental results are shown in Fig 4.

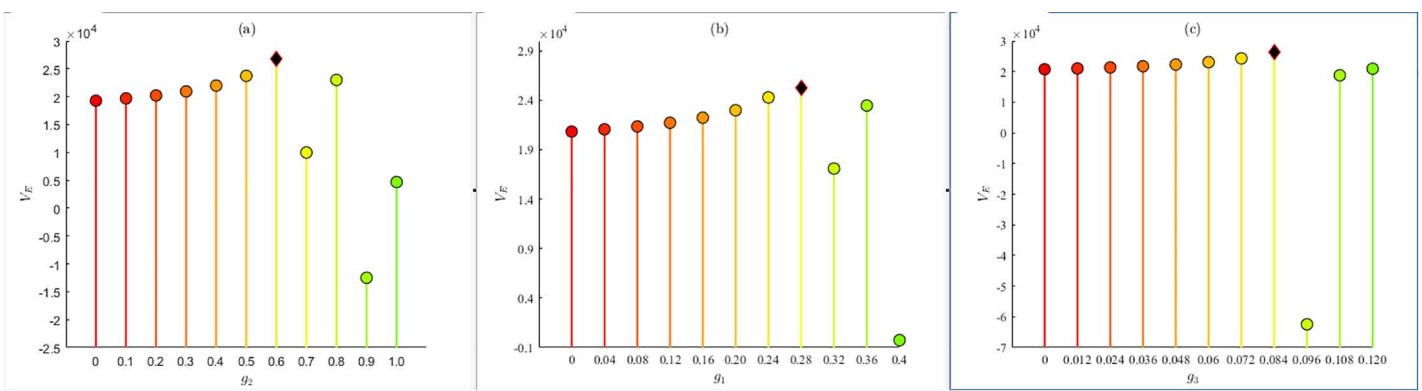

**Fig 3. The relationship between the ratio of crisis response resources and enterprises revenue.**

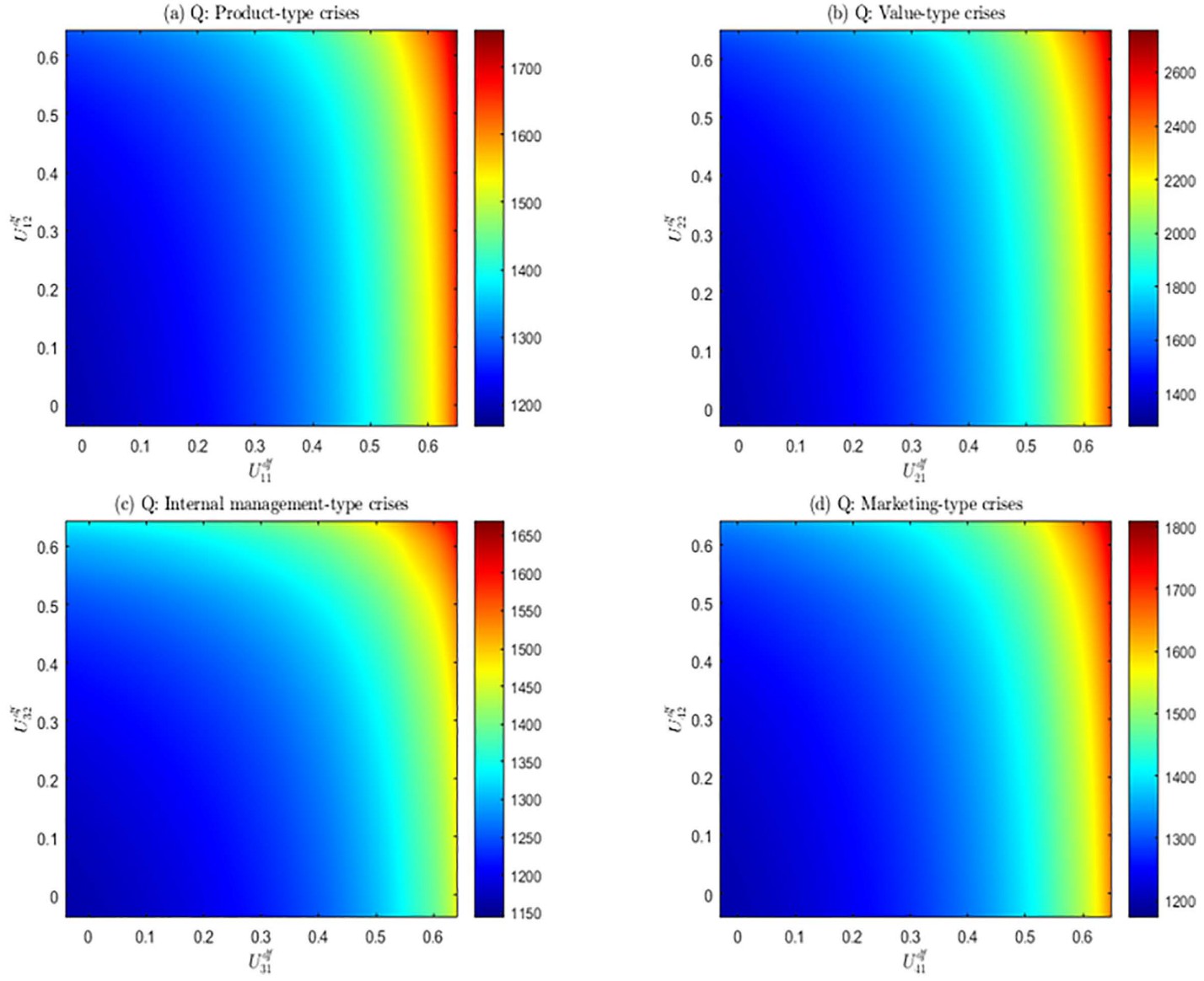

**Fig 4. The relationship between the subsidy coefficient ($U_{i1}^{df}$ and $U_{i2}^{df}$) and the spreading range of public opinion (Q).**

Fig 4 shows the relationship between the two kinds of subsidy coefficients and the spreading range of public opinion ($Q$) for the four types of crises. Overall, the spreading range of public opinion derived from various crises expands with the increase of $U_{i1}^{df}$ and $U_{i2}^{df}$, that is, the subsidy coefficient of enterprises to media and netizens has a significant positive effect on the spreading scope of public opinion. Increasing the enterprises subsidy coefficient effectively share the investment costs of media and netizens, and reduce their public opinion spreading costs. This will then encourage the media and netizens' public opinion spreading behaviors and assist enterprises in guiding public opinion. The experimental results are in line with expectations.

In addition, it was found that the spreading scope of public opinion also have different reactions to the changes in $U_1$ and $U_2$. Compared with $U_2$, $Q$ is more sensitive to the changes of $U_1$. Especially in Fig 4(a–b), $Q$ shows obvious change

along $U_1$'s direction, but there is almost no significant change along $U_2$'s direction. The above-mentioned phenomenon can be attributed to the following aspects. Media's reporting on social platforms can be regarded as goal-oriented professional behaviors, playing a pivotal role in the spreading process of public opinion, and the content they released can quickly break through the limitations of the circle and form a viral spread. Meanwhile, media holds the power to screen, process and authoritatively interpret information, serving as the main source of information for netizens. Netizens' spreading is more of a spontaneous behavior driven by emotions, and their ability to release and spread information is very limited. And the spreading scope and social influence of information decline exponentially. In addition, netizens tend to trust and spread public opinion released by highly credible media. Therefore, appropriately tilting subsidies for media can also promote the efforts of netizens to spread information, thereby maximizing the propagation scope of public opinion beneficial to enterprises.

It can be known from the above analysis that the spreading scope of public opinion derived from crisis is more dependent on media's reporting behavior. When facing different types of crises, while not neglecting the subsidy for netizens, enterprises should appropriately tilt their resources towards to media. With the help of their high coverage and strong influence, it can actively guide public opinion and assist enterprises get through crisis smoothly.

## 6. Conclusions

Under the new media environment, the diversity and complexity of crises result in varied impacts on enterprises. The selection of appropriate crisis guidance strategy is crucial in guiding the evolution trend of public opinion derived from different types of crises, especially facing the limited enterprises governance resources. This paper integrated differential game theory into enterprises public opinion guidance, built differentiated decision-making scenarios based on management practices and solved the optimal equilibrium under different guidance strategies for various kinds of crises and their corresponding effectiveness from a theoretical perspective. Then, numerical simulations were introduced to identify key factors influencing the evolution and guidance of public opinion, providing targeted measures for enterprises to guide public opinion derived from crisis. The main findings are shown as follows.

1) In the scenario without guidance, media and netizens have the lowest degree of effort, and with the intervention of enterprises guidance, their efforts have improved. **2)** Compared with decentralized decision-making scenario, media and netizens have higher degree of effort and enterprises pay less. **3)** The optimization of guiding effect of public opinion can be achieved by adjusting the cooperation form between media and netizens (i.e., ***xj***), or by updating the incentive strategy (i.e., ***df***). The implicit guidance coefficient plays a key role when comparing the strategic choices of media and netizens in ***xj*** and ***df*** scenarios. **4)** When facing different types of crises simultaneously, enterprises should prioritize the values-type crises, followed by the product-type crises, marketing-type crises, and internal management-type crises, and the allocation ratio of four types of crises is approximately 60%, 28%, 8.4%, and 3.6% respectively, which can achieve optimal overall benefit. **5)** The spreading scope of public opinion was affected by the subsidy coefficient of enterprises to media and netizens positively at the same time, but compared with netizens, it is more sensitive to changes in media's subsidy coefficient. Based on research findings, this paper provided a series of suggestions to help enterprises actively guide public opinion and get through crises smoothly.

The findings of this paper provide insights for enterprises to effectively guide public opinion derived from crisis. However, decision-making during crisis is a complex management issue. The interrelationships among different types of crises, crisis prediction and the recurrence of crisis leading to diminishing marginal utility will significantly impact the guiding effect of public opinion derived from crisis. These areas require further in-depth research in the future.

## Supporting information

**S1 Data. Data file.**

(XLSX)

## Acknowledgments

We would like to express our sincere appreciation to the anonymous referees for their valuable comments and suggestions.

## Author contributions

**Data curation:** Jiakun Wang, Huiying Chen.

**Methodology:** Jiakun Wang, Xiaotong Guo.

**Validation:** Jiakun Wang, Yawei Li.

**Visualization:** Jiakun Wang, Yawei Li, Xiaotong Guo.

**Writing – original draft:** Jiakun Wang, Xiaotong Guo.

**Writing – review & editing:** Jiakun Wang, Yawei Li, Huiying Chen.

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
