## [Decision Letter · Decision Letter 0]

9 Jul 2025

Dear Dr. Wang,

Thank you for submitting your manuscript to PLOS ONE. After careful consideration, we feel that it has merit but does not fully meet PLOS ONE’s publication criteria as it currently stands. Therefore, we invite you to submit a revised version of the manuscript that addresses the points raised during the review process.

We look forward to receiving your revised manuscript.

Kind regards,

Giorgio Rizzini

Guest Editor

PLOS ONE

Journal Requirements: 

2. You indicated that ethical approval was not necessary for your study. We understand that the framework for ethical oversight requirements for studies of this type may differ depending on the setting and we would appreciate some further clarification regarding your research. Could you please provide further details on why your study is exempt from the need for approval and confirmation from your institutional review board or research ethics committee (e.g., in the form of a letter or email correspondence) that ethics review was not necessary for this study? Please include a copy of the correspondence as an ""Other"" file.

4. Please note that funding information should not appear in any section or other areas of your manuscript. We will only publish funding information present in the Funding Statement section of the online submission form. Please remove any funding-related text from the manuscript.

 “Natural Science Foundation of Shandong Province (ZR2024MG049; ZR2021QG035) and Shandong Social Science Planning and Research Project (22DGLJ27)” 

6. We note that your Data Availability Statement is currently as follows: All relevant data are within the manuscript and in Supporting Information files.

See the Reviewers' comments.

Reviewers' comments:

Reviewer's Responses to Questions

**Comments to the Author**

1. Is the manuscript technically sound, and do the data support the conclusions?

Reviewer #1: Yes

Reviewer #2: Partly

2. Has the statistical analysis been performed appropriately and rigorously?

Reviewer #1: N/A

Reviewer #2: N/A

3. Have the authors made all data underlying the findings in their manuscript fully available?

Reviewer #1: Yes

Reviewer #2: No

4. Is the manuscript presented in an intelligible fashion and written in standard English?

Reviewer #1: Yes

Reviewer #2: No

Reviewer #1: The manuscript, titled "Research on enterprise network public opinion guiding decision-making considering crisis differentiation" provides an interesting view and methodology to describe different crisis of confidence in given institutions that have public perception playing an important role to their future evolution. The manuscript is sound and covers all relevant scenarios to evaluate such application with the presented simulations.

The first suggestion that I'd like to make to the authors would be to discuss in more depth the main choice of parameters in section 5 given by the mentioned 17 experts, for overall clarity and interpretability of the current results. Moreover, it would be interesting to see a discussion focused on the changes to the dynamics of each scenario with different values set to these parameters.

Also, even having well drawn conclusions from numeric simulations, and the parameters used having a been chosen by field experts, the manuscript extend those results directly to recommendations to real enterprises, when no real data was used to support these claims. I believe that it would suffice to reformulate the text to take that into consideration, and leave only suggestions that such conclusions can be applied to real world scenarios.

Lastly, I think figures could benefit from larger font sizes, to improve readability. Figure 4 specifically should also set the z axis in the same scale for all panels, that way allowing for a better comparison between variables.

Overall, future research with this methodology could attempt to study recurring crisis of confidence to a given enterprise, and how public opinion would be affected by these repeating events, which in turn would help understand the behavior of decade-spawning institutions and also allow for a study of stability of those systems.

Reviewer #2: Overview

This paper investigates how enterprises can effectively steer public opinion during crisis situations. Leveraging a differential game decision model, the authors analyze the strategic interactions between companies, media, and netizens across various crisis types—namely product, values, internal management, and marketing crises.

The study identifies that public opinion management is most effective when firms employ a dual guidance strategy (df in the authors’ code), combining explicit incentives for media and netizens with implicit guidance mechanisms. This approach maximizes the overall benefits for the company by fostering a virtuous cycle of positive information diffusion and engagement. The paper’s simulations reveal that centralized decision-making yields greater benefits than decentralized approaches, though it is more challenging to implement in practice.

The authors recommend an optimal resource allocation according to the impact of each crisis type, with 60% directed to value crises, 28% to product crises, 8.4% to marketing crises, and 3.6% to internal management crises. The effectiveness of public opinion steering is shown to be also highly sensitive to the subsidy ratio provided to the media, underscoring the need for targeted investment in media guidance.

Assessment

The manuscript examines an interesting problem applying known optimal control techniques based on cooperative and noncooperative differential games and drawing on previous work (not always accessible due to the language, such as D. Wu, Y. Yang. Study on the differential game model for supply chain with consumers’ low carbon preference, Chinese Journal of Management Science. vol. 29, no. 4, pp. 126-137, Apr. 2021).

Overall, the results appear sound, even though the proofs of the presented propositions are completely lacking. I understand that such proofs basically involve computations, but perhaps including the calculations in the appendix would be helpful for the reader, at least in the case of one proposition (e.g., the first one, case (f)). In general, the manuscript has limitations and flaws and cannot be published in its current form. It requires significant revisions. Below, I provide some comments on possible revisions and improvements.

Comments

1. A thorough revision of the language is necessary. Some sentences are unclear, and there are many typos. For example in the Literature Review: ‘the mechanism of public opinion affect mechanism, […]’ , or at the end of Sec. 3: ‘Enterprises provide relevant guidance and subsidy measures to media to encourage them to spread truthful and favourable information to enterprises to clarify the truth.’ Or, again, in Sec. 4.1: ‘The Hamilton-Jacobi-Bellman (HJB)

equation is used to solve the model, drawing on the approach outlined in.’(?).

2. Some acronyms are never defined throughout the paper, for example, OSN, SNP, WP and many others.

3. The crisis classification is clear and well-described. However, the model section requires a complete revision. In particular, the "Basic Assumption" section is merely a list of definitions of the quantities involved in the differential game. It should be rewritten as a section that properly illustrates the basic assumptions, not as a mere list pf quantities. generally, indexes on quantities are generally so small that they are difficult to read. Some parameters are also never defined, such as the eta_M and eta_N.

4. I understand that C_Mi(t), that is the cost of reporting by the media at the time of the type i crisis, is a quadratic function in the corresponding effort degree. This is a common assumption indeed. So I can understand what the cost for media is for producing and spread information but I don’t understand what kind of cost is involved for netizens, who mainly are users of social networks, I suppose. Could the author please better clarify what kind of cost they are referring to?

5. The authors propose a differential equation (1) for the credibility of the media. Is this equation found in the literature, and why was it chosen? Is it universal? I would also like to emphasize the role and meaning of the omega coefficient, which represents implicit guidance, as well as the difference between unguided and explicitly guided types on the left side of Eq. (1).

6. The p discount rate is defined at the end of Sec. 3 but it appears only below in the HJB Eqs (5) first.

7. Before Section 4.1, I suggest clarifying the classification of crises and strategies because there are four types of crises and four types of guidance strategies. A table or explanatory scheme would be useful to avoid any confusion.

8. Fix the name of the author Jorgensena in the text and bibliography.

9. In Eq. (5) and the similar equations that follow, I would explicitly state the variables. What variables does the maximum apply to?

10. In Eq. (7) I don’t understand the dependence on time t of the right hand side. Is there no integral over time?

11. The comments for each type of strategy in Sec 4.1, and followings, limit to the positive or negative correlations between the optimal decision variables and other parameters. I would expect some more interpretation of the optimal values.

12. In Corollary 2 they appear for the first time these terms for omega: stealthy bootstrapping coefficient and the invisible bootstrap coefficient.

13. In Sec. 5, the authors justify the selection of parameter values based on the opinions of several experts. I personally have no reason not to trust their expertise, but it would be nice to find some real motivation for the values of at least some of the involved parameters.

14. The captions of the figures must be improved. For example, the meaning of the four curves in Fig. 2 should be explained. Moreover, the readability of the legends in the figures should be increased.

15. In Sec. 5.3, please clarify why the authors only determine g₂ and then assign the other gi values according to their priority. What does this mean? Below, I see three figures for three of them. How are these figures obtained?

**Do you want your identity to be public for this peer review?** For information about this choice, including consent withdrawal, please see our Privacy Policy

Reviewer #1: No

Reviewer #2: No

---

## [Author Response · Author response to Decision Letter 1]

11 Sep 2025

Please see the attachment named "Response to Editor & Reviewers".

---

## [Decision Letter · Decision Letter 1]

23 Oct 2025

Dear Dr. Wang,

Thank you for submitting your manuscript to PLOS ONE. After careful consideration, we feel that it has merit but does not fully meet PLOS ONE’s publication criteria as it currently stands. Therefore, we invite you to submit a revised version of the manuscript that addresses the points raised during the review process.

We look forward to receiving your revised manuscript.

Kind regards,

Giorgio Rizzini

Guest Editor

PLOS ONE

Journal Requirements:

Reviewers' comments:

Reviewer's Responses to Questions

**Comments to the Author**

Reviewer #1: All comments have been addressed

Reviewer #2: All comments have been addressed

2. Is the manuscript technically sound, and do the data support the conclusions?

Reviewer #1: Yes

Reviewer #2: Yes

3. Has the statistical analysis been performed appropriately and rigorously?

Reviewer #1: N/A

Reviewer #2: N/A

4. Have the authors made all data underlying the findings in their manuscript fully available?

Reviewer #1: Yes

Reviewer #2: Yes

5. Is the manuscript presented in an intelligible fashion and written in standard English?

Reviewer #1: Yes

Reviewer #2: Yes

Reviewer #1: The manuscript, titled "Research on enterprise network public opinion guiding decision-making considering crisis differentiation" provides an interesting view and methodology to describe different crisis of confidence in given institutions that have public perception playing an important role to their future evolution. This new revision vastly improves on the original methodology, covering multiple flaws from the first version and also clarifying both the formula's fundamentals and the assumption made by authors to the modeling processes. All the questions and suggestions I have made for the first version have been answered, and I feel that this manuscript is almost publication-ready as of now.

I just want to point out that the caption of Fig. 1. c) might be a typo. Shouldn't it be "credibility of the netizens", as it was in the original manuscript? I've personally not found other similar disparities while reading this revised version.

Reviewer #2: The authors have carried out a thorough revision of the first version of the manuscript. They have also responded in a very detailed and comprehensive manner to all the questions and comments raised.

**Do you want your identity to be public for this peer review?** For information about this choice, including consent withdrawal, please see our Privacy Policy

Reviewer #1: No

Reviewer #2: No

---

## [Author Response · Author response to Decision Letter 2]

25 Oct 2025

Please see the attachment named "Response to Editor & Reviewers".

---

## [Editor Report · Decision Letter 2]

28 Oct 2025

Research on enterprise network public opinion guiding decision-making considering crisis differentiation

PONE-D-25-19210R2

Dear Dr. Wang,

We’re pleased to inform you that your manuscript has been judged scientifically suitable for publication and will be formally accepted for publication once it meets all outstanding technical requirements.

Kind regards,

Giorgio Rizzini

Guest Editor

PLOS ONE

---

## [Editor Report · Acceptance letter]

PONE-D-25-19210R2

PLOS ONE

Dear Dr. Wang,

I'm pleased to inform you that your manuscript has been deemed suitable for publication in PLOS ONE. Congratulations! Your manuscript is now being handed over to our production team.

Kind regards,

on behalf of

Dr. Giorgio Rizzini

Guest Editor

PLOS ONE